# Short-Term Effect of the Combined Application of Rice Husk Biochar and Organic and Inorganic Fertilizers on Radish Growth and Nitrogen Use Efficiency

**DOI:** 10.3390/plants13172376

**Published:** 2024-08-26

**Authors:** War War Mon, Hideto Ueno

**Affiliations:** 1Department of Bioresource Production Science, United Graduate School of Agriculture, Ehime University, 3-5-7 Tarumi, Matsuyama 790-8566, Ehime, Japan; 8warwarmonyau@gmail.com; 2Department of Agro-Biological Science, Graduate School of Agriculture, Ehime University, 3-5-7 Tarumi, Matsuyama 790-8566, Ehime, Japan

**Keywords:** rice husk biochar, radish, organic fertilizer, inorganic fertilizer, nitrogen use efficiency

## Abstract

Research on soil biochar fertilization has mainly been conducted on cereal crops, and information on its potential for radish production remains inconsistent. Therefore, a pot experiment was conducted to examine the short-term effects of rice husk biochar on radish growth and nitrogen use efficiency (NUE). An investigation was conducted with two application rates of biochar alone, (10 t ha^−1^ (B10) and 25 t ha^−1^ (B25), and biochar + chicken manure application with and without NPK fertilizer. The results indicated that the application of biochar 25 t ha^−1^ + chicken manure (B25:CHM) and the combination of biochar 25 t ha^−1^ + chicken manure + NPK fertilizer (B25:CHM:NPK) significantly increased root yield by improving NUE, fertilizer recovery efficiency (RE_N_), agronomic efficiency (AE), nitrogen harvest index (NHI), and retaining soil NH_4_^+^-N. Although biochar application alone did not significantly influence radish growth on a short-term basis, B10 and B25 increased root yields by 10% and 20%, respectively, compared with the control. Notably, the role of biochar application when combined with organic and inorganic fertilizers was to retain fertilizer N and promote N uptake efficiency by radishes, as higher rates of biochar resulted in higher NUE. Our results suggest that B25:CHM is a suitable combination for organic farming.

## 1. Introduction

With concerns about climate change, extreme weather conditions, and the lack of arable land, crop production must be safeguarded to mitigate potential food crises [1]. Despite the importance of crop production, soil organic matter (SOM) in most agricultural soils is estimated to be depleted [2]. Although nitrogen (N) is an important nutrient and essential element for plant growth, it is not sustainable to apply synthetic N fertilizers to soils with physical degradation and low fertility because it can cause acidification and nutrient imbalances [3]. If N fertilizers are applied far beyond the demands of crops, this results in low N-use efficiency and large nitrate residues in the soil [4]. Therefore, restitution of soil organic matter (SOM) by increasing N-use efficiency is important for sustainable crop production, particularly in low-fertility soils.

Amendment of soil with biochar can serve as a tool for storing soil organic carbon because biochar is a carbon-rich material [5]. The conversion of wastes such as rice husk into biochar facilitates energy production and recycles wastes, thereby improving sustainable waste management [6,7]. Interest in biochar application is widespread and stems from observations of old agricultural soils in the Amazon Basin, called Terra Preta. Compared to neighboring natural soils, these soils are much more fertile and contain a higher level of carbon [8]. These observations led to the pyrolysis of a wide range of feedstocks at different temperatures under oxygen-limited conditions to produce biochar [9,10]. This manufacturing process produces biochar with desirable properties, such as a high surface area, high porosity, rich functional groups, and high cation exchange capacity (CEC) [11]. Due to these unique properties of biochar, it would be an eco-friendly solution focusing on environmental sustainability to address the problem of soil contamination, organic pollutants, and greenhouse gas emissions [12,13].

Currently, the combined application of biochar and organic fertilizers has given rise to a growing interest in agricultural production [14]. Livestock manure is one of the most effective substitutes for synthetic fertilizers [15]. They can be applied either directly (fresh) or after composting [16]. Generally, composting reduces some chemical and biological risks related to organic amendments [17]. Stable compost products made from organic wastes are rich in humus, which is beneficial for organic fertilizers [18]. Organic fertilizers can improve the biological, physical, and chemical properties of soil, as well as provide beneficial effects on agriculture [19]. Among organic fertilizers, chicken manure is commonly used in agriculture and is a nutrient-rich organic waste containing considerable amounts of N, phosphorus, and potassium [20]. In addition, a previous study indicated that biochar and N fertilizer had significant interactions at higher yields with increasing rates of biochar application with NPK fertilizer [21]. Thus, the combined application of biochar and either organic or inorganic fertilizer is expected to enhance soil fertility and provide nutrients to short-season crops with high nutrient requirements, such as radish.

Radish (*Raphanus sativus* L.) is grown and consumed worldwide, and many colors are available (red, purple, black, yellow, and white through pink). Radishes are typically consumed raw as crunchy vegetables, primarily in salad [22]. It is generally rich in carbohydrates, sugars, dietary fibers, proteins, fat, and fluoride. They contain water-soluble vitamins and minerals. Therefore, it has been recognized as beneficial to human health [23]. Moreover, radish is a good option for small-scale farmers because they pay back in approximately 30 days [24]. Radish is not only a short-cycle crop but also a highly nutrient-demanding crop. Therefore, the co-application of fertilizers with organic amendments, such as biochar, is required to avoid rapid nutrient losses [25]. In addition, more research is needed to fully understand the effect of the application of biochar separately or together with different nitrogen sources on the nitrogen use efficiency of crops [26]. To date, research on soil biochar fertilization has mainly been conducted on cereal crops [14,27]. The nutritional effects of biochar on short-cycle crops are inconsistent. Sousa et al. [24] reported that biochar application alone can supply nutrients in a short period of time, resulting in increased radish productivity. In contrast, some results indicated that the expected benefit of biochar alone without poultry manure may not be achieved within the first year of radish cultivation [28]. To address this research gap, we conducted a pot experiment to examine the effect of biochar in sole and its combined effect with organic and inorganic fertilizers on radish yield, N-use efficiency, and soil chemical properties on a short-term basis.

## 2. Results

### 2.1. Plant Growth of Radish under Various Treatments

Our experiments involved measuring radish growth using several parameters such as plant height, number of leaves, and leaf color. As shown in Figure 1a, plant height varied from 3.02 cm to 18.7 cm throughout the growth stage. Plant height increased steadily from 7 days after sowing (DAS), and then B10:CHM, B25:CHM, B10:CHM:NPK, and B25:CHM:NPK increased sharply at 21 DAS. Among the various treatments, B10: CHM: NPK and B25: CHM: NPK showed significant heights at 28 DAS, whereas B25 showed the lowest height, similar to that of C and B10. The number of leaves increased from 7 to 28 DAS under various treatments, except for the C, B10, and B25 treatments, which decreased at 28 DAS (Figure 1b). The highest number of leaves was recorded for B10:CHM:NPK (7.6), and the lowest number was recorded for C (2.6). Regarding chlorophyll content, the SPAD value varied from 13.3 to 36.3 during the entire growth period. As shown in Figure 1c, there was a decreasing trend, which gradually decreased from 7 to 21 DAS and then increased at 28 DAS; however, the differences among the treatments were not significant.

### 2.2. Germination and Radish Yield Response to Different Treatments

The germination rates and radish yields are compared in Table 1. Seed germination began on the third day after sowing, and the highest germination rate was observed in the B25:CHM:NPK treatment, which induced 100% germination. The lowest germination rates were recorded for B10. No significant differences in germination rate were detected across treatments. Compared to C, B10 reduced the germination rate by 0.41%, whereas B25, B10:CHM, B25:CHM, and B10:CHM:NPK increased the germination rate by 27.3%, and B25:CHM:NPK increased the germination rate by 36.4%. The data revealed that there was a significant effect of the different treatments on both the radish yield and dry biomass of leaves and roots. The maximum leaf yield was observed in B10:CHM:NPK, followed by B25:CHM:NPK, B25:CHM, B10:CHM, B25, B10, and C. However, B10:CHM:NPK and B25:CHM:NPK were not significantly different in leaf yield. The lowest leaf yield was obtained with C; however, it was not significantly different from that of B10 and B25. The same trend was observed for leaf dry biomass weight (B10:CHM:NPK > B25:CHM:NPK > B25:CHM > B10:CHM > B25 > B10 > C). The data regarding root and dry biomass yields showed that B25:CHM:NPK produced the maximum yield, and the minimum yield was noted in C. We found the same trend for both root yield and root dry biomass, as follows: B25:CHM:NPK > B25:CHM > B10:CHM:NPK > B10:CHM > B25 > B10 > C. Biochar application alone (B10 and B25) did not show a significant effect on root and dry biomass yields compared to C. However, B10 increased 10% and B25 increased 20% in root yield, whereas B10 increased 6.3% and B25 increased 12.5% in dry root biomass compared with C.

### 2.3. Nitrogen Use Efficiency (NUE) under Different Treatments

Table 2 shows the NUE values. NUE, RE_N_, and AE increased for B25:CHM:NPK but the differences were not statistically significant compared to B25:CHM. The highest PE and NHI values were obtained for B25:CHM. B10 and B25 showed very low values for all NUE indices.

### 2.4. Standardized Regression Coefficients

The standardized regression coefficients derived from the treatment factors are presented in Table 3. A standardized regression coefficient was used to compare the effects of biochar, chicken manure, and NPK fertilizer on the explanatory variables. NPK fertilizer had the greatest effect on radish leaf yield, whereas chicken manure contributed to increased root yield, NUE, RE, PE, AE, and NHI. Biochar tended to increase radish root yield more than leaf yield. The NPK fertilizer reduced the PE and NHI.

### 2.5. Soil Chemical Properties after Cultivation

The chemical properties of the soil after the experiment are listed in Table 4. Although the values of soil pH ranged from 7.3 to 7.8, treatments with biochar tended to increase the soil pH compared to the control. The pH values of B25 and B10:CHM increased by 4.1%, B25:CHM increased by 6.8%, B10:CHM:NPK increased by 5.5%, and B25:CHM:NPK increased by 1.4% relative to that of C. The highest pH (7.8) was observed for B25:CHM. EC value varied in the range of 12.5 to 187.5 (µS/cm). B25:CHM:NPK exhibited the highest EC value. EC values decreased in the following order: B25:CHM:NPK > B10:CHM:NPK > B25:CHM > B10:CHM > B25 > B10 > C. The highest soil total N was recorded in B25:CHM:NPK; however, it did not differ significantly from that of B10:CHM, B25:CHM, and B10:CHM:NPK. The order of magnitude was arranged from highest to lowest as follows: B25:CHM:NPK, B10:CHM:NPK, B25:CHM, B10:CHM, C, B25, and B10.

In this study, soil NH_4_^+^-N increased by 2.4% in B10, 3.1% in B25, 61.7% in B10:CHM, 109.4% in B25:CHM, 122.2% in B10:CHM:NPK, and 148.2% in B25:CHM:NPK relative to C. The soil ammonium content in B25:CHM:NPK greatly increased, but was not significantly different from that in B10:CHM:NPK and B25:CHM. There was no significant difference between biochar application alone (for both 10 t ha^−1^ and 25 t ha^−1^) and the control for ammonium N. The order of magnitude of soil NH_4_^+^-N was B25:CHM:NPK > B10:CHM:NPK > B25:CHM > B10:CHM > B25 > B10 > C. Regarding soil NO_3_^-^-N, there was no significant difference among the treatments with biochar; however, it was statistically significant relative to the control. The highest nitrate N content was recorded for B10:CHM:NPK.

After harvest, the value of available P ranged between 241.1 mg kg^−1^ and 674.4 mg kg^−1^ with the order of B25:CHM:NPK > B10:CHM:NPK > B25:CHM > B10:CHM > B25 > B10 > C. Although non-significant differences were detected in soil available P among C, B10, B25, and B10:CHM, B10 increased by 7.2%, B25 increased by 11.0%, and B10:CHM increased by 28.8% compared with C. Regarding soil-exchangeable K, the value ranged between 0.09 mg kg^−1^ and 0.42 mg kg^−1^. Application of B25:CHM:NPK and B25:CHM influenced soil potassium, and the order of magnitude was arranged from highest to lowest: B25:CHM:NPK > B25:CHM > B10:CHM:NPK > B10:CHM > B25 > B10 > C. The range of soil-exchangeable magnesium concentration was 1.12 mg kg^−1^ to 7.20 mg kg^−1^. A relatively high concentration of Mg was observed in B10:CHM:NPK. Statistical analysis revealed no significant differences in soil-exchangeable Mg among B25:CHM:NPK, B10:CHM:NPK, B25:CHM, and B10:CHM. The application of the different treatments did not change the soil Ca content after cultivation. However, there was an increase of 13.3%, 7,8%, 16.4%, 10.1%, 18.0%, and 11.7% in B10, B25, B10:CHM, B25:CHM, B10:CHM:NPK, and B25:CHM:NPK relative to C. Based on these results, the order of soil Ca concentrations was as follows: B25:CHM > B10:CHM:NPK > B10:CHM > B10 > B25:CHM:NPK > B25 > C.

## 3. Discussion

### 3.1. Response of Plant Growth to Different Biochar Treatments

Plant growth characteristics such as plant height, number of leaves, and SPAD value responded positively to the combined application of biochar and chicken manure with or without NPK fertilizer, indicating that B10:CHM:NPK and B25:CHM:NPK showed significantly higher plant height. There is evidence to suggest that the combined application of biochar and organic or NPK fertilizers can improve soil quality and plant performance [29,30]. All applications of biochar+ organic + inorganic fertilizer could supply the required nutrients because they would contribute to a better nutritional environment during the active vegetative period, thus leading to increased cell elongation and multiplication, ultimately enhancing plant height. According to our findings, B10:CHM:NPK produced the highest number of leaves (Figure 1b). It should be noted that B10:CHM:NPK did not reach a significance level compared to B25:CHM:NPK and B25:CHM. B10:CHM:NPK increased leaf number more than B25:CHM:NPK and B25:CHM because readily available forms of NPK were positively reflected in radish leaf development, regardless of the biochar application rate.

SPAD values showed a declining trend until 21 DAS. At 28 DAT, the SPAD increased again with the application of B10, B25, B10:CHM, B25:CHM, B10:CHM:NPK, and B25:CHM:NPK. The B10:CHM:NPK and B25:CHM:NPK treatments showed the highest SPAD values from 14 to 28 DAS (Figure 1c). The application of rice husk biochar alone cannot supply sufficient N because of its low N content. These results are in accordance with those of [31], who explained that a low level of N in biochar led to a decrease in the SPAD value. In addition, N released from the applied fertilizers would be taken up by radish roots; thus, transportation to radish leaves would be small and SPAD values would decrease until 21 DAS. Zhang et al. [32] demonstrated that N, P, and K nutrients could be removed by the fleshy root of radish.

### 3.2. Radish Germination, Yield, and Nitrogen Use Efficiency under Different Treatments

Although biochar had no obvious effect on seed germination, increasing the application rate of biochar combined with organic and inorganic fertilizers (B25:CHM:NPK) resulted in the highest radish germination percentage (Table 1). This result corresponds to that of [33], who reported that peanut hull biochar did not significantly influence barley germination. Solaiman et al. [34] investigated the effects of biochar type and rate on the germination rate and early growth of wheat. In both the soilless Petri dishes and soil-based bioassays, the type and application rate of biochar affected wheat seed germination and seedling growth. Ali et al. [35] demonstrated that corncob biochar could be mixed with soil at different rates (0.5%, 1%, 1.5%, 2%, 2.5%, and 3% *w*/*w*). They confirmed that higher application rates have a neutral to favorable impact on maize seed germination and seedling development.

Maximum radish leaf yield was obtained using B10:CHM:NPK. According to the regression coefficient results (Table 3), NPK fertilizer greatly increased radish leaf yield, indicating that radish leaves obtained nutrients mainly from NPK fertilizer. This result is consistent with the number of leaves. Regarding root yield, B25:CHM, B10:CHM:NPK, and B25:CHM:NPK application increased radish root yield, and chicken manure had a greater effect on root yield (Table 1 and Table 3). The lowest root yield was found in C because neither organic nor inorganic fertilizers were applied to the control treatment. In addition, the experimental soil had very low fertility, with a low total N of 0.02%, and radish growth under C depended only on the existing soil N. Therefore, the radish plants under the C treatment could not obtain sufficient required N, which led to a decreased yield. The variations in leaf or root dry biomass yield under different treatments were consistent with the variations in radish yield (Table 1). This result is consistent with that of [20], who reported that crop yield is directly correlated with biomass.

Our findings showed that the maximum fertilizer NUE was achieved with the application of B25:CHM:NPK, followed by B25:CHM and B10:CHM:NPK (Table 2). Standardized regression analysis revealed that chicken manure significantly influenced NUE. Notably, a higher application rate of biochar combined with chicken manure, with or without NPK fertilizer, significantly increased NUE compared to biochar alone. The combined application of biochar along with organic or inorganic fertilizers stimulates plant growth and increases fertilizer utilization owing to the easily degradable components of organic and inorganic fertilizers [36,37]. Additionally, biochar can retain NO_3_^-^-N and NH_4_^+^-N in manure [28]. As previously stated, the higher surface area of biochar makes it more effective at binding nutrients and improving NUE [38,39]. In addition, biochar can facilitate electron transfer to soil-denitrifying microorganisms, which accelerates the last step of denitrification, providing complete denitrification (N_2_), and thus reducing N loss as N_2_O [40]. Despite the positive effects of biochar on soil, its relatively low nutrient composition and resistance to biodegradation depend on the production temperature [41,42]. Therefore, biochar decomposition is likely to be minimal, and N immobilization probably occurs over a short time. In this study, biochar itself had a low total N content of 0.81%; thus, biochar application alone could not improve NUE.

B25:CHM:NPK had the highest RE_N_, followed by B10:CHM:NPK, and B25:CHM. This indicates that the higher the N input, the higher the RE_N_. Considering the results of our study, both chicken manure and NPK fertilizer can influence RE_N_ (Table 3) by supporting the required N uptake by radishes, thereby improving recovery efficiency (Table 2). The highest NHI value was obtained for B25:CHM, followed by B25:CHM:NPK. As mentioned above, the high level of N accumulated in radish roots was mainly derived from chicken manure, and biochar could enhance NUE; therefore, the radish plant’s N translocation to yield could be the highest under the B25:CHM treatment. This combination resulted in the highest PE. Among the treatments assessed in this study, B25:CHM is a potential amendment for organic farmers.

### 3.3. Effect of Biochar Application alone and Combined with Organic and Inorganic Fertilizer on Soil Chemical Properties after the Experiment

As a result of this experiment, rice husk biochar application combined with chicken manure, either with or without chemical fertilizer, improved soil chemical properties even over a short period of time compared to the control (see Table 4). Soil pH can directly influence the availability of nutrients because too low to too high a soil pH leads to the insolubility of some nutrients and limits their availability. The various treatments had no significant effects on soil pH because the pH of the soil used in this experiment was higher than that of the biochar. Even though there was no significant difference among the treatments, the soil pH value in the presence of biochar tended to increase slightly relative to the control. Generally, biochar application can increase the soil pH because of its alkaline nature. Basic cations in biochar (Ca^2+^, Mg^2+^, and K^+^) can be converted into alkaline substances (oxides, hydroxides, and carbonates) through pyrolysis. When these components dissolve, biochar acts as a liming ingredient, increasing the soil pH [43].

In agriculture, the soil electrical conductivity (EC) is an indirect indicator of available plant nutrients and salinity. In the present study, soil amendments with the B25:CHM:NPK treatment resulted in a significantly higher soil EC (Table 4). Notably, soils treated with biochar had significantly increased soil EC compared to the control. This is a positive result because our experimental soil had a very low EC with a value of 21 μS/cm. Soils with low EC tend to lose nutrients easily because of their poor capacity to hold cations. Soil EC is directly proportional to nutrient concentration [44]. Rice husk biochar had an EC of 856.3 µS cm^−1^ and contributed to an increase in soil EC after cultivation. The presence of oxidized functional groups, ash, and alkali compounds, such as Na, K, Mg, and Ca, can influence the EC value of biochar [45]. Zou et al. [46] reported that the release of these soluble compounds was positively correlated with an increase in soil EC. Additionally, adding manure improves organic matter, which can provide a pool of nutrients and ions that can be released into the soil solution, thereby increasing EC [47]. In addition, the obtained results are in agreement with those of Bhatt et al. [48], who indicated that organic fertilizer along with NPK fertilizer enhanced the EC of the soil because the breakdown of organic fertilizers released electrolytes.

After harvest, the highest total soil N content was observed in the B25:CHM:NPK treatment, which was significantly higher than that in the B10, B5, and C. However, B25:CHM:NPK was not significantly different from B10:CHM, B25:CHM, and B10:CHM:NPK (Table 4). Chicken manure, which contains 4.05% of the total N, can lead to increased soil N. Application of B25:CHM:NPK, B10:CHM:NPK, and B25:CHM significantly improved soil ammonium N compared to the control (Table 4). The order of magnitude of soil ammonium N was almost the same as that of soil total N. Soil ammonium N, rather than nitrate N, may have greatly affected plant growth in the present study. Based on the raw materials and the process of manufacturing biochar, the available forms of N vary significantly. Some studies have reported that owing to its porous nature and high surface area, biochar can retain N, thereby increasing the amount of N in the soil [49,50]. In other words, N released from chicken manure and NPK fertilizer was retained on the biochar surface, which led to increased soil available N when combined with biochar, chicken manure, and NPK fertilizer.

Nguyen et al. [51] indicated that soil amendment with biochar decreased soil NH_4_^+^-N and NO_3_^-^-N levels by an average of 11% and 10%, respectively. According to their results, biochar produced from carbohydrates at low treatment temperatures and hydrothermal conditions reduced soil inorganic N more than the other biochars. It is possible that the high surface area (adsorption) and high C/N ratio of the biochar caused the immobilization. Biochar application alone may reduce soil N for a short period, but the available N may increase over time [51,52]. Therefore, organic manure with a low C/N ratio is a good option for promoting biochar production. Mineralization may be rapid and release nutrients from the organic manure [28]. Biochar may create space for soil microorganisms to reproduce and enhance the nutrients released from organic manure [53,54]. Furthermore, inorganic fertilizers can release nutrients in a short time because of their readily available characteristics; hence, the combination of biochar, chicken manure, and NPK fertilizer would be a perfect combination to promote soil available N, even in a short time.

Based on the results of this study, applications of B10, B25, and B10:CHM were not significant in soil-available P content compared with C. Even though we applied the same application rate of chicken manure (0.01 g P_2_O_5_ pot^−1^) and NPK fertilizer (0.25 g P_2_O_5_ pot^−1^) in both B10:CHM:NPK and B25:CHM:NPK, B25:CHM:NPK showed the highest statistically significant result. This indicates that a higher application rate of biochar can retain more available soil P. Biochar can serve as a growth medium for P-solubilizing bacteria owing to its high specific surface area, high internal porosity, and capacity to adsorb organic molecules [55]. B25:CHM:NPK had the highest soil-exchangeable K value, followed by B25:CHM. There is no doubt that the high K content in biochar and chicken manure can contribute to an increase in soil-exchangeable K. Moreover, all combined applications of biochar and chicken manure, with and without NPK fertilizer, enhanced exchangeable Mg. Such enhancement may be due to the relatively high pH of the soil, as well as the ash content in biochar, which could lead to greater qualities of exchangeable basic cations such as K, Mg, and Ca. Furthermore, poultry manure contains high concentrations of basic cations that can serve as liming materials. Therefore, when these cations are released, the cation content and ionic strength of the soil increase, thereby increasing the exchangeable ions [56]. There was no significant difference in soil-exchangeable Ca among the treatments because the experimental soil had relatively high Ca, which led to an abundance of soil-exchangeable Ca, even in the control.

## 4. Materials and Methods

### 4.1. Experimental Site, Treatments, and Radish Cultivation

A precise pot experiment was conducted in a greenhouse (33.83° N, 132.79° E) at Ehime University in Matsuyama City, Japan. This study was conducted during the cropping season of 3 June to 5 July 2023. The soil was collected from a mountainous area in Toon City, Ehime Prefecture, Japan. The basic properties of the experimental soil are listed in Table 5.

The rice husk biochar and chicken manure were thoroughly mixed with 4 kg of air-dried soil. Then, well-mixed soil was filled into Wagner pots with an area of 0.02 m^2^ one week before sowing the radish seeds. NPK fertilizer was applied on sowing day with the application rate of 1.6 g pot^−1^ of NPK fertilizer. All the pots were placed on the blocks to prevent radiation and latent heat exposure. The experiment consisted of five replicates in a completely randomized design. The experiment included seven treatments: (i) control (C, no amendment), (ii) B10 = 10 t ha^−1^ of biochar (20 g pot^−1^), (iii) B25 = 25 t ha^−1^ of biochar (50 g pot^−1^), (iv) B10:CHM = 10 t ha^−1^ of biochar (20 g pot^−1^) + 5 t ha^−1^ of chicken manure (10 g pot^−1^), (v) B25:CHM = 25 t ha^−1^ of biochar (50 g pot^−1^) + 5 t ha^−1^ of chicken manure (10 g pot^−1^), (vi) B10:CHM:NPK = 10 t ha^−1^ of biochar (20 g pot^−1^) + 5 t ha^−1^ of chicken manure (10 g pot^−1^ )+ 120 kg ha^−1^ of NPK (15-15-15), (vii) B25:CHM:NPK = 25 t ha^−1^ of biochar (50 g pot^−1^) + 5 t ha^−1^ of chicken manure (10 g pot^−1^) + 120 kg ha^−1^ of NPK (15-15-15). Commercially available rice husk biochar and chicken manure were used in this study. Table 6 lists the chemical characteristics of rice husk biochar. Chicken manure had the following properties: total N 4.05%, total C 25.04%, C/N 6.19, and available P 1334 mg kg^−1^.

In this study, radishes (*Raphanus sativus* L. var. radicula Pers.) were used. Three seeds were sown in each pot. Germination tests were performed one week after sowing. The seedlings were thinned to one plant per pot. The germination percentage was calculated based on Fedeli et al. [57]
(1)$$Germination(\%)=\frac{\mathrm{Number}\;\mathrm{of}\;germinated\;seeds\;per\;pot}{Total\;number\;of\;seeds\;planted\;per\;pot}\times 100$$

Irrigation with pure water was performed at least once a week, depending on the weather conditions, to maintain soil moisture throughout the root zone. No supplemental fertilizer was applied during the experiment.

### 4.2. Plant Growth, Radish Yield Measurements, and Nitrogen Use Efficiency

The plant height, number of leaves, and SPAD values were recorded weekly. A meter ruler was used to measure the plant height. The chlorophyll content of fully healthy and fully expanded leaves was measured from the bottom, center, and top of the leaves per plant (average of three points measured) using a SPAD 502-Plus chlorophyll meter (Konica Minolta, Inc., Osaka, Japan). The radishes were harvested at commercial maturity, 31 days after sowing. Plants were carefully uprooted to avoid damaging their roots, and excess soil around the roots was washed with water. The leaves were then separated from the tubers, and the fresh weights of the leaves and roots per pot were recorded using a digital balance. After weighing, the leaves and roots were oven dried at 70 °C. When they reached a constant weight, the dry matter weights of leaves and roots were recorded. Radish leaf and root yields were determined by weight per plant.

The fertilizer N recovery efficiency (RE_N_), physiological efficiency (PE), and agronomic efficiency (AE) were calculated based on Congreve et al. [58]. PlantN, which is the plant N content, was calculated as the sum of N concentration in both radish leaves and roots multiplied by their dry biomass, whereas PlantN(control) was the N concentration under the control treatment multiplied by its dry biomass. Fertilizer N was calculated as the total N concentration derived from the applied fertilizer.

Nitrogen use efficiency (NUE) was defined as the ratio of the radish root yield to the total N derived from the applied fertilizer. NUE was determined by using the equation proposed by Alkharabsheh et al. [59].
(2)$$NUE\;(g\;g^{-1}\;N)=\frac{Yield}{Fertilizer}$$

The fertilizer N recovery efficiency (RE_N_) is an indicator of fertilizer N uptake by radish plants. It was computed by subtracting the plant N content under the control treatment (PlantN (control)) from the plant N content (PlantN) under different treatments.
(3)$$REN\;(\%)=\frac{PlantN-PlantN(control)}{Fertilizer\;N}$$

The amount of N supplied by fertilizers that plant tissues contribute to yield is known as physiological efficiency (PE). PE was calculated by using the following equation:(4)$$PE\;(g\;g^{-1}\;N)=\frac{Yield-Yield(control)}{PlantN-PlantN(control)}$$

Agronomic efficiency (AE) indicates how N fertilizer contributes to the yield in comparison to a control.
(5)$$AE\;(g\;g^{-1}\;N)=\frac{Yield-Yield\left(control\right)}{FertilizerN}$$

The N harvest index (NHI) was calculated based on Fageria et al. [60] as the N accumulated in the radish roots divided by the total N content in the radish leaves and roots. The yield N was computed as the N concentration in the radish roots multiplied by the dry biomass.
(6)$$NHI(\%)=\frac{yield\;N}{PlantN}\times 100$$

### 4.3. Measurement of Soil Chemical Properties

All soil chemical properties were determined as described by Mon et al. [14]. To determine how the chemical properties of the soil changed, soil analysis was performed before and after cultivation. Prior to examination, the soil samples were air dried and sieved (≤2 mm). A pH meter (B-212, HORIBA, Kyoto, Japan) was used to measure the pH of the soil from soil–water suspensions (1:2.5, *v*/*v*), and an EC meter (Horiba Twin Cond Conductivity meter B173) was used to measure the EC of the soil. This study employed calorimetric methods to assess the amounts of ammonium N (NH_4_^+^-N) and nitrate N (NO_3_^-^-N) in the soil. Available P was measured using the Troug technique. Atomic absorption spectrometry was used to quantify the quantities of exchangeable K, Ca, and Mg after extraction using 1 M ammonium acetate. The dry combustion method was used to determine the total C and N contents using a Vario Max CN elemental analyzer (Elementar Analysensysteme GmbH, Langenselbold, Germany).

### 4.4. Statistical Analysis

We performed a one-way analysis of variance (ANOVA) using the Real Statistics Resource Pack for all data. Tukey’s HSD test at a significance level of *p* < 0.05 was used to analyze the differences in mean values. Multiple regression analysis was performed by standardizing all variables to obtain standardized regression coefficients.

## 5. Conclusions

It was found that B25:CHM:NPK produced better results in improving NUE, plant growth, and yield than the other treatments. An important finding of this study was that NPK fertilizer greatly increased radish leaf yield, whereas chicken manure had a greater effect on radish root yield. In addition, B25:CHM and B25:CHM:NPK had no significant effect on soil NH_4_^+^-N, NO_3_^-^-N, exchangeable K, Mg, Ca, or root yield. Our experiments indicated that B25:CHM is a suitable combination for organic farmers. No significant differences were found between treatments in terms of soil pH, soil NO_3_^-^-N, seed germination, or SPAD. The application of biochar alone (B10 and B25) did not significantly influence on radish growth. However, B10 increased by 10% and B25 increased 20% in radish root yield relative to the control. Notably, when biochar is combined with organic and inorganic fertilizers, a higher application rate of biochar can contribute to greater fertilizer N uptake efficiency by radishes in a short time. Since biochar ages with time, more research is needed to explore the changes in biochar mechanisms in soil over time and to investigate the effects of aging on crop production and nitrogen use efficiency.

## Figures and Tables

**Figure 1 plants-13-02376-f001:**
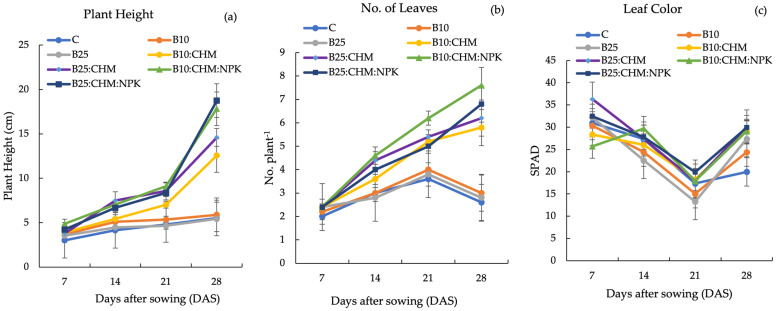
Plant height (**a**), number of leaves (**b**), and leaf color (**c**) under different treatments. Data are presented as the mean ± standard error. C = control, B10 = 10 t ha^−1^ of biochar application; B25 = 25 t ha^−1^ of biochar application; B10:CHM = 10 t ha^−1^ of biochar + 5 t ha^−1^ of chicken manure; B25:CHM = 25 t ha^−1^ of biochar + 5 t ha^−1^ of chicken manure; B10:CHM:NPK = 10 t ha^−1^ of biochar + 5 t ha^−1^ of chicken manure + NPK; B25:CHM:NPK = 25 t ha^−1^ of biochar + 5 t ha^−1^ of chicken manure + NPK.

**Table 1 plants-13-02376-t001:** Application of biochar alone and in combination with organic and inorganic fertilizers: its effects on radish germination and yield component.

Treatments	Germination (%)	Radish Yield (g pot^−1^)		Dry Biomass Yield (g pot^−1^)	
		Leaves	Root	Leaves	Root
C	73.3 ± 6.67 a	0.20 ± 0.03 de	0.10 ± 0.03 bcd	0.06 ± 0.004 de	0.016 ± 0.002 bcd
B10	73.0 ± 12.5 a	0.22 ± 0.04 e	0.11 ± 0.02 d	0.07 ± 0.01 e	0.017 ± 0.004 d
B25	93.3 ± 6.67 a	0.23 ± 0.05 d	0.12 ± 0.02 c	0.08 ± 0.01 d	0.018 ± 0.006 c
B10:CHM	93.3 ± 6.67 a	3.00 ± 0.26 c	2.11 ± 0.43 b	0.33 ± 0.05 c	1.02 ± 0.110 b
B25:CHM	93.3 ± 6.67 a	4.11 ± 0.25 b	7.38 ± 0.75 a	0.46 ± 0.03 b	2.09 ± 0.325 a
B10:CHM:NPK	93.3 ± 6.67 a	8.09 ± 0.35 a	5.03 ± 0.78 a	0.91 ± 0.03 a	1.46 ± 0.370 a
B25:CHM:NPK	100 ± 0.00 a	7.97 ± 0.64 a	9.86 ± 1.96 a	0.82 ± 0.08 a	2.28 ± 0.545 a
*p* value	0.08	<0.001	<0.001	<0.001	<0.001

Data are presented as mean ± standard error. Significant differences (*p* < 0.05) are indicated by different letters in the same column among the treatments. C, control; B10 = 10 t ha^−1^ biochar application, B25 = 25 t ha^−1^ biochar application; B10:CHM = 10 t ha^−1^ of biochar + 5 t ha^−1^ of chicken manure; B25:CHM = 25 t ha^−1^ biochar + 5 t ha^−1^ of chicken manure; B10:CHM:NPK = 10 t ha^−1^ of biochar + 5 t ha^−1^ of chicken manure + NPK; B25:CHM:NPK = 25 t ha^−1^ biochar + 5 t ha^−1^ of chicken manure + NPK.

**Table 2 plants-13-02376-t002:** Nitrogen use efficiency under different treatments.

Treatments	NUE (g g^−1^ N)	RE_N_ (%)	PE (g g^−1^ N)	AE (g g^−1^ N)	NHI (%)
B10	0.60 ± 0.15 c	0.5 ± 0.36 d	0.03 ± 0.01 d	0.009 ± 0.15 c	13.1 ± 3.19 b
B25	0.24 ± 0.04 d	0.6 ± 0.24 c	0.05 ± 0.06 c	0.006 ± 0.04 d	9.5 ± 2.49 c
B10:CHM	3.70 ± 0.76 b	7.5 ± 1.38 b	0.50 ± 0.08 ab	3.54 ± 0.76 b	53.5 ± 6.11 a
B25:CHM	9.11 ± 0.93 a	11.6 ± 1.49 a	0.93 ± 0.09 a	8.99 ± 0.93 a	67.0 ± 3.76 a
B10:CHM:NPK	8.38 ± 1.30 a	17.3 ± 2.35 a	0.55 ± 0.02 b	8.23 ± 1.30 a	48.4 ± 7.91 a
B25:CHM:NPK	11.7 ± 2.33 a	18.5 ± 2.31 a	0.75 ± 0.12 a	11.63 ± 2.33 a	56.5 ± 7.17 a
*P value*	<0.001	<0.001	<0.001	<0.001	<0.001

Data are presented as mean ± standard error. Significant differences (*p* < 0.05) are indicated by different letters in the same column among the treatments. C, control; B10 = 10 t ha^−1^ biochar application, B25 = 25 t ha^−1^ biochar application; B10:CHM = 10 t ha^−1^ of biochar + 5 t ha^−1^ of chicken manure; B25:CHM = 25 t ha^−1^ biochar + 5 t ha^−1^ of chicken manure; B10:CHM:NPK = 10 t ha^−1^ of biochar + 5 t ha^−1^ of chicken manure + NPK; B25:CHM:NPK = 25 t ha^−1^ biochar + 5 t ha^−1^ of chicken manure + NPK. NUE: fertilizer nitrogen utilization efficiency; RE_N_: fertilizer nitrogen recovery efficiency; PE: physiological efficiency; AE: agronomic efficiency; NHI: nitrogen harvest index.

**Table 3 plants-13-02376-t003:** Standardized regression coefficients derived from treatment factors based on multiple regression analysis.

Response Variables		Explanatory Variables					
	Leaf Yield	Root Yield	NUE	RE	PE	AE	NHI
Biochar	0.04	0.32 *	0.28 *	0.12	0.31 **	0.28 **	0.13
Chicken Manure	0.49 ***	0.47 ***	0.57 ***	0.56 ***	0.91 ***	0.58 ***	0.99 ***
NPK Fertilizer	0.61 ***	0.30 *	0.35 **	0.52 ***	−0.08	0.34 **	−0.16

NUE: fertilizer nitrogen utilization efficiency; RE_N_: fertilizer nitrogen recovery efficiency; PE: physiological efficiency; AE: agronomic efficiency; NHI: nitrogen harvest index. * Significant at *p* < 0.05. **: significant at *p* < 0.01, ***: significant at *p* < 0.001.

**Table 4 plants-13-02376-t004:** Chemical properties of low-fertility soil after harvest.

Treatments	pH	EC (µS cm^−1^)	Total N (%)	NH_4_-N (mg kg^−1^)	NO_3_-N (mg kg^−1^)	Available P (mg kg^−1^)	Ex. K (cmol_(c)_ kg^−1^)	Ex. Mg (cmol_(c)_ kg^−1^)	Ex. Ca (cmol_(c)_ kg^−1^)
C	7.3 ± 0.19 ab	12.5 ± 2.2f	0.016 ± 0.011 b	7.11 ± 0.8 cd	1.28 ± 0.1 b	241.1 ± 5.2 cd	0.09 ± 0.02 d	1.12 ± 0.1 d	12.8 ± 0.7 ab
B10	7.3 ± 0.05 ab	20.5 ± 2.3e	0.009 ± 0.003 d	7.28 ± 0.6 d	2.15 ± 0.4 a	258.5 ± 20.0 cd	0.18 ± 0.01 c	2.02 ± 0.5 c	14.5 ± 0.1 ab
B25	7.6 ± 0.04 a	41.5 ± 5.6d	0.015 ± 0.002 c	7.33 ± 0.4 c	2.04 ± 0.6 a	267.6 ± 15.2 d	0.23 ± 0.01 c	2.70 ± 0.5 b	13.8 ± 1.0 b
B10:CHM	7.6 ± 0.04 a	118.5 ± 4.3c	0.025 ± 0.005 a	11.50 ± 1.2 b	2.70 ± 0.4 a	310.5 ± 10.8 c	0.20 ± 0.01 ab	4.82 ± 0.3 a	14.9 ± 0.3 a
B25:CHM	7.8 ± 0.05 a	131.5 ± 6.3b	0.033 ± 0.008 a	14.89 ± 1.0 a	3.44 ± 0.5 a	439.9 ± 28.1 b	0.32 ± 0.06 a	6.61 ± 0.5 a	17.1 ± 0.4 a
B10:CHM:NPK	7.7 ± 0.01 a	140.5 ± 5.5b	0.034 ± 0.002 a	15.80 ± 1.6 a	4.22 ± 0.5 a	525.8 ± 28.4 b	0.26 ± 0.002 b	7.20 ± 1.6 a	15.1 ± 0.2 a
B25:CHM:NPK	7.4 ± 0.03 b	187.5 ± 4.1a	0.050 ± 0.001 a	17.65 ± 1.1 a	3.89 ± 0.9 a	674.4 ± 30.8 a	0.42 ± 0.03 a	6.35 ± 0.6 a	14.3 ± 0.9 a
*p* value	<0.001	<0.001	<0.001	<0.001	<0.05	<0.001	<0.001	<0.001	<0.001

Data are presented as means ± standard error. Significant differences (*p* < 0.05) between treatments are indicated by different letters in the same column. Ex. = exchangeable; C = control; B10 = 10 t ha^−1^ biochar application; B25 = 25 t ha^−1^ biochar application; B10:CHM = 10 t ha^−1^ biochar + 5 t ha^−1^ chicken manure; B25:CHM = 25 t ha^−1^ biochar + 5 t ha^−1^ chicken manure; B10:CHM:NPK = 10 t ha^−1^ biochar + 5 t ha^−1^ chicken manure + NPK; B25:CHM:NPK = 25 t ha^−1^ biochar + 5 t ha^−1^ chicken manure + NPK.

**Table 5 plants-13-02376-t005:** Characteristics of experimental soil.

Measurement	Units	
Total N	%	0.02
Total C	%	0.03
C/N		1.68
Available NH_4_^+^-N	mg kg^−1^	11.7
Available NO_3_^-^-N	mg kg^−1^	0.23
Available Phosphorus	mg kg^−1^	125.3
Exchangeable K content	cmol_(c)_ kg^−1^	0.006
Exchangeable Ca content	cmol_(c)_ kg^−1^	9.84
Exchangeable Mg content	cmol_(c)_ kg^−1^	0.88
pH		7.86
EC	μS cm^−1^	21

**Table 6 plants-13-02376-t006:** Chemical properties of rice husk biochar.

Measurements	Unit	
Pyrolysis temperature	°C	900–1000
pH		6.45
EC	(µS cm^−1^)	856.3
CEC	(cmol_(c)_ kg^−1^)	25.4
Exchangeable K	(mg kg^−1^)	14960
Exchangeable Mg	(mg kg^−1^)	421.1
Exchangeable Ca	(mg kg^−1^)	2415
Ash	%	44.9
Volatile Matter	%	18.7
Total N	%	0.81
Total C	%	33.7
C/N		41.8

## Data Availability

Data are contained within the article.

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
