# Peer review of "Short-Term Effect of the Combined Application of Rice Husk Biochar and Organic and Inorganic Fertilizers on Radish Growth and Nitrogen Use Efficiency"

_plants, 2024, doi:10.3390/plants13172376_

Round 1

Reviewer 1 Report

Comments and Suggestions for Authors

The manuscript entitled ‘’ Short-term effect of sole and combined application of rice-husk-biochar, organic, and inorganic fertilizers on growth of radish and nitrogen use efficiency in low-fertility soil’’ investigated short-term effects of different rice-husk-biochar on radish growth and nitrogen use efficiency (NUE). Please see the comments as below:

Abstract

The abstract needs some changes. Rewrite the results with percentages (how % increase or decrease?). Please add suggestion for future research (in one or two lines). Keywords should differ from the title, please specify them.

Lime 14-15: Please add also the pot application rate (gr kg-1),

Line 17: One space is missing, manure+ NPK -- > manure + NPK

Introduction

Lines 42-43: One special and critical thing about biochar production is limited oxygen during biochar production, please add it. You can also use this reference for improve this section:

https://doi.org/10.1016/j.enconman.2023.117924

Lines 45-47: it’s not related to only rice husk biochar, all types of biochars can improve soil quality, and enhance plant growth.

The innovation of the study should be clear at the end of this section. The background of the research is not enough. There are some literatures near to the current research, so the authors should compare the current research with previous ones and bold the gap of knowledge of rice husk biochar. Please add some previous research here and improve the introduction. You can use this research https://doi.org/10.3390/agronomy12092106

 Results and discussion

The main problem of this section is related to the state of results. Most of the results are descriptive. Increases and decreases are visible in figures. The point is that how much is this increase or decrease in percent? Treatments should be compared with control by percentage.

Please check significant letters and numbers of Tables 1, 2, and 4 (all of them) should be checked, for example in Table 1, standard error B10:CHM:NPK, or significant letters of Germination % and …

Or in Table 4, pH for B10 and B25:CHM:NPK is 7.4, but significant letters are different!

Materials and Methods

Add longitude and latitude of the experimental site.

Why did the author select this high pyrolysis temperature (900-1000 °C) for rice husk biochar?! Is there any specific reason?

Please add also the pot application rate for all treatments (gr kg-1)

Lines 387-400: please add the reference for each equation and also refer them in the text.

Conclusion

It is better if the authors add suggestions for future research at the end of the conclusion.

References

There aren’t any 2024 references in your paper, I recommend use some update reference in the discussion and introduction section!!

Author Response

Dear Reviewer 1,

We extend our sincere gratitude to the anonymous reviewers for your invaluable feedback and constructive criticism. Your insightful comments and suggestions have significantly improved the quality and clarity of our manuscript. We appreciate the time and expertise they dedicated to evaluating our work, which has undoubtedly enhanced its scientific rigor and potential impact in the field. Please see the file. Thank you.

Reviewer 2 Report

Comments and Suggestions for Authors

Dear authors,

i read the manuscript with interest. In the attached PDF you can find some comments for increasing the level of the manuscript. Main criticism are releated to the introduction and M&M section, results and discussion very good.

I will be very happy to review the manuscript after corrections since I believe there is potential in the manuscript submitted by the authors.

Author Response

Dear Reviewer 2,

We extend our sincere gratitude to the anonymous reviewers for your invaluable feedback and constructive criticism. Your insightful comments and suggestions have significantly improved the quality and clarity of our manuscript. We appreciate the time and expertise they dedicated to evaluating our work, which has undoubtedly enhanced its scientific rigor and potential impact in the field. Please see the file. Thank you.

Round 2

Reviewer 1 Report

Comments and Suggestions for Authors

The authors incorporated all the comments carefully and the revised version is acceptable. 

Reviewer 2 Report

Comments and Suggestions for Authors

Manuscript can be accepted, congratulations to the authors